# Curing Assessment of Concrete with Hyperspectral Imaging

**DOI:** 10.3390/ma14143848

**Published:** 2021-07-09

**Authors:** Lisa Ptacek, Alfred Strauss, Barbara Hinterstoisser, Andreas Zitek

**Affiliations:** 1Department of Civil Engineering and Natural Hazards, University of Natural Resources and Life Sciences, 1190 Vienna, Austria; alfred.strauss@boku.ac.at; 2Department of Material Sciences and Process Engineering, University of Natural Resources and Life Sciences, 1190 Vienna, Austria; barbara.hinterstoisser@boku.ac.at; 3FFoQSI—Austrian Competence Centre for Feed and Food Quality, Safety & Innovation, FFoQSI GmbH, Technopark 1D, 3430 Tulln, Austria; andreas.zitek@boku.ac.at

**Keywords:** concrete curing, near infrared spectral signatures, hyperspectral imaging, condition assessment, non-destructive testing (NDT), k-means clustering

## Abstract

The curing of concrete significantly influences the hydration process and its strength development. Inadequate curing leads to a loss of quality and has a negative effect on the durability of the concrete. Usually, the effects are not noticed until years later, when the first damage to the structure occurs because of the poor concrete quality. This paper presents a non-destructive measurement method for the determination of the curing quality of young concrete. Hyperspectral imaging in the near infrared is a contactless method that provides information about material properties in an electromagnetic wavelength range that cannot be seen with the human eye. Laboratory tests were carried out with samples with three different curing types at the age of 1, 7, and 27 days. The results showed that differences in the near infrared spectral signatures can be determined depending on the age of the concrete and the type of curing. The data was classified and analyzed by evaluating the results using k-means clustering. This method showed a high level of reliability for the differentiation between the different curing types and concrete ages. A recommendation for hyperspectral measurement and the evaluation of the curing quality of concrete could be made.

## 1. Introduction

The curing of concrete is largely responsible for the quality of concrete components. To achieve durability, concrete must be treated and protected. Professional curing focuses on the chemical processes during the hydration which is associated with the hardening of concrete. The hardening process and compaction of concrete starts immediately after its pouring, e.g., in its formwork. During this phase, the chemical hydration generates the cement stone, which ties together the grains of the composite material concrete [1]. At this stage it is of the utmost importance for a proper hydration to have enough moisture available to develop the desired concrete strength. An insufficient moisture range on the concrete surface causes a drying process associated with the development of capillary pores, and in consequence, to an imperfect hydration and reduced strength [2,3]. As a result, the structure in the peripheral concrete zone remains porous, which leads to lower surface strength, reduced weather resistance, lower resistance to chemical attack, the development of early shrinkage cracks, etc. [4,5,6].

In order to avoid or minimize these negative processes for the lifespan of a reinforced concrete component, the following harmful processes are minimized for young concrete:premature drying of the surfacestrong temperature fluctuationsand excessive cooling or heating [7]

The primary goal of a good curing is therefore the creation of a dense concrete structure, avoiding capillary pores and cracks reaching from the surface to the reinforcement layer. A good curing can be obtained with the following measures:by protecting the concrete surface with foils and protection mats (in this case the concrete curing is additionally favored due to removing constraints from the structure, for example, when concreting floor slabs.)by moistening the concrete surface with a water mistby applying a liquid protective film or after-treatment agentby leaving the concrete in the formwork [8,9,10]

Checking or monitoring the correct curing of concrete components is not an easy task under construction conditions. Processes are required that are easily manageable, can be used over a wide area, are reproducible, and allow unambiguous statements. Among other things, the test procedures must not destroy or impair the properties of the concrete. A promising method that has already found its way into several evaluation processes is hyperspectral imaging. It has already been used in individual cases to check the developmental strength of concrete.

Hyperspectral imaging is a non-destructive method for the determination of material properties. It provides data of reflectance information over a certain wavelength range in the electromagnetic spectrum that cannot be seen by the human eye, e.g., in the case of the near infrared. According to [11,12], the deterioration of concrete regarding chloride ingress or carbonation can be detected by using spectroscopy. Concrete quality assessment with spectral imaging had already been investigated in [13], where concrete specimens with different w/c ratios (water-to-cement ratio) were classified for density prediction. Moreover, in [14], concrete specimens with varying w/c ratios and curing times could be differentiated according to their concrete strength. The effects of different construction practices on the spectral characteristics of concrete were investigated in [15]. Concrete treatments with better characteristics could be clearly separated from concrete treatments with poor properties. In Table 1, an overview of the methodologies and the procedures for assessing the quality and deterioration of concrete from previous hyperspectral investigations based on literature research is shown.

Previous investigations showed that different concrete qualities can be determined with spectral imaging. The aim of this work was to evaluate the potential of near-infrared hyperspectral imaging for assessing the quality of curing for sufficient concrete hydration.

In particular, it should be clarified: (a) whether an incorrect hydration process, and consequently, a non-optimal strength development can be recognized; (b) at which times the application can be used optimally; (c) if the spectral signature of the hyperspectral analysis differs dependent on the curing type; and (d) how the signatures change over the time of an optimally cured, a less well cured, and a poorly cured concrete.

## 2. Materials and Methods

### 2.1. Instrumentation and Measuring Principle

Concrete samples were scanned using a NIR hyperspectral imaging instrument (Zeutec Opto-Elektronik GmbH, Rendsburg, Germany) located at the University of Natural Resources and Life Sciences, Department of Material Sciences and Process Engineering, Institute of Physics and Materials Science, in Vienna, Austria with improved hard- and software components. The actual system components of the instrument are:Xenics NIR camera (Xeva-USBFPA-1.7-320-TE1-100 Hz camera with InGaAs focal plane array sensor with 2% pixel noise–XEVA 6179; 0.9–1.7 μm; 320 × 256 pixel matrix; 12 bit);Specim N17E spectrograph operating in the range of 950–1650 nm with an IR optimized objective lens;600 mm Y-table gear;stable diffuse 45/0 illumination created by halogen bulbs

The experimental setup used for the hyperspectral analyses is shown in Figure 1. The surface of the samples was scanned in lines (“pushbroom imaging”). The scanned samples were moved further during the measurement using a stepper motor mounted on a y-table to cover the entire surface of the sample. A 3D image was recorded with spatial data in the x and y directions, as well as spectral information in the z direction [17]. Each pixel of the resulting image contained reflectance information in a wavelength range from about 950 to 1650 nm. The continuous display of the reflectance intensity in the measured wavelength range is referred to as the spectral signature [18] and was evaluated individually for each recorded pixel.

The whole system was controlled using Argus software (Version 2010-2016 for University of Natural Resources and Life Sciences Vienna - BOKU, F. Firtha, Budapest, Hungary) [19]. Preparing for the measurement of the samples, a roundness test measurement was made to check if the pixel sizes from the field of view were calculated correctly. A two-point calibration was performed by considering a white panel as a reflectance reference and a black measurement (by covering the objective lens with its cover) as an absorbance reference before scanning. Reflectance values were obtained by dividing the reflectance of the sample by the reflectance of the white panel.

The y-length needed to be defined in the Argus software according to the size of the sample. The final setup in the Argus software is shown in Figure 2. The reflectance, in a wavelength range of 959 to 1631 nm, was scanned with a spectral resolution of 3.33 nm.

### 2.2. Sample Preparation

Test series were carried out on concrete samples with composition and properties as shown in Table 2. The dimensions of the samples (10 cm × 10 cm × 10 cm) were chosen due to the limited conditions given by the setup of the measurement system (larger objects can be measured by modification of the setup or by using, e.g., portable devices). The samples were treated with different curing conditions for the first seven days after concreting (see Table 3). The samples with curing type NB1 were cured under good conditions at 20 °C and wrapped in foil. The NB2 samples were not treated at all and were also stored at 20 °C. The NB3 samples underwent poor curing conditions with no protection of the surface, and were stored at 30 °C and lower relative humidity than samples NB1 and NB2. After seven days, all samples were stored under the same conditions (20 °C and no protection of the surface). In total, four samples per curing type were produced and examined.

As indicated in Table 3, different strengths were achieved for the NB1, 2, and 3 concrete samples after 28 days of hardening. The values show that the curing quality had an influence on the strength development; the worse the curing conditions, the lower the strength of the concrete. Additionally, the concrete samples as listed in Table 3 were tested in a project [20] with other non-destructive methods, such as gas permeability and ultrasonic testing. They showed a similar trend and confirmed the decrease in the quality of concrete with inadequately performed curing.

Five faces per sample (the labelled face was excluded from the measurements) were measured with the hyperspectral camera on the first day (day 1), one week (day 7), and 4 weeks (day 27) after concreting (day 0). The measurement data were subsequently processed and compared with one another to determine differences in the spectral signatures with respect to unequally treated concrete samples and the various days of measurement.

The data from the samples of the same curing type and age all showed similar results. In the following, the results of three representative measurements per curing type and measurement day are used to compare the data.

### 2.3. Data Analysis

The entire hyperspectral imaging workflow, from sample preparation to data analysis and evaluation, is shown in Figure 3. In the following, the procedures for data analysis (starting from point 5 in Figure 3) and processing are explained in detail.

The obtained hyperspectral data in the form of a spectral hypercube consisting of spatial and spectral information were treated with a noise reduction algorithm to remove the so called “salt and pepper” noise, which detects unserviceable bright or dark pixels. These pixels were replaced by a neighboring pixel algorithm script in MATLAB^®^ (R2017b, The MathWorks, Inc., Natic, MA, USA) [21] that detects x and y pixels in an x, y, z matrix that are significantly different from their neighbors on the z axis [22].

The data were further processed with the data analysis software PLS-Toolbox in Matlab. The region of interest of each image was defined by selecting the concrete surface and removing the background in the picture.

The spectral pre-processing of the data was done by normalization and Savitzky–Golay smoothing (SGS) [23] with PLS-Toolbox. By choosing the 1st derivative order at SGS, unambiguously results could be achieved. In one case of the analyzed images, the 2nd derivate order had to be selected to get clear results.

Data analysis was carried out using k-means clustering. K-means clustering is an unsupervised data classification algorithm that divides the dataset into different groups. First, k cluster centers, called centroids, are randomly assigned, and each data point (in this case the individual pixels of the recordings) is assigned to the cluster with the closest mean value [24,25,26]. Afterwards, the centroids are recalculated from the current clusters. Each data point is then assigned to the cluster closest to it, and so on. This is repeated until the value of the centroid does not change any more [27,28,29]. The number of clusters is defined in advance. In order to obtain meaningful results, it is necessary in some cases to adjust the number of clusters and to repeat the calculation.

K-means clustering was used to determine whether samples with different curing and results from different measurement days could be differentiated from one another by means of pixel classification. Hence, images of the three different curing types and images with different concrete ages were merged to one single image for statistical analysis. Different combinations of images, as shown in Table 4, were arranged for data processing.

Combinations 1–3 included measurements of all three curing types. Images of NB1, NB2, and NB3 were compared on day 1, day 7, and day 27 after concreting (see Table 4, combination 1–3). Three measured images of each curing type were included in each combination. Hence, the merged pictures consisted of a total of nine images, with three images for each curing type.

A similar arrangement was made to find out if there was a change in the data by comparing measurement day 1, day 7, and day 27 (see Table 4, combination 4–6). The measurements of the three mentioned days were compared to one another for each kind of curing (NB1, NB2, and NB3). Again, three images per day and curing type were used for comparison over time.

In Figure 4, two examples of arrangements of the merged images are given. Picture (**a**) shows the images of the three curing types on measurement day 1, and (**b**) shows the composition of images from NB 1 samples from day 1 to day 27. The merged images are displayed in false colors, based on the average reflectance value of each pixel. The other four combinations were created in the same way.

The optimal number of classes for k-means clustering resulted from the interpretability of the associated spectral signatures. In general, clear signature differences could be achieved with three clusters (in a few cases with four clusters), where the explanatory clusters contained relevant spectral information and the additional clusters contained only a comparably small number of pixels, representing artefact values near holes and edges or caused by direct reflection.

Each pixel of a merged image was assigned to one of the three (or four) clusters, later referred to as classes. After k-means clustering, the pixels were shown in the colors red, blue, and green, depending on their class affiliation. Each class had its own characteristic spectral signature. The proportions of pixels per class for the respective samples or on the respective measurement day allowed conclusions to be drawn about the material properties of the various curing types or the hydration progress.

The statistics showed that pixels assigned to classes A and B were especially predominant for all analyses. Class C and class D (only with the choice of four classes) contained only a few pixels and therefore were not representative. They were not considered for further evaluation of the data.

Two characteristic absorption bands around 1340 nm and 1400 nm could be defined in the spectral signatures for classes A and B. The extent of absorption differed depending on the curing type and the measurement day. The water band was found in the wavelength range around 1400 nm. The absorption in this area was strongest in class A, which means that the pixels in class A had a higher water content compared to class B.

In general, classes A, B, and C could be distinguished as follows:Class A: spectral signatures showed the strongest absorption at 1400 nm compared to class B. A significant proportion of pixels was assigned to class A.Class B: spectral signatures showed less absorption at 1400 nm compared to class A. A significant proportion of pixels were assigned to class B.Class C (and D): The proportion of pixels was not representative, mainly representing artefact pixels near holes and at the edges.

## 3. Results

The results of the analyses of all image combinations are shown in Figure 5, Figure 6, Figure 7, Figure 8, Figure 9 and Figure 10. For each combination the classes were evaluated. They differed in the properties of their spectral signatures and in the pixel proportions of the images.

The results of combinations 4–6 are shown in Figure 8, Figure 9 and Figure 10. These combinations show the development of data of the NB1 (Figure 8), NB2 (Figure 9), and NB3 (Figure 10) samples from day 1 to day 27. With all three curing types, a clear change in the class distribution (b) could be determined over the duration of hydration. The greatest differences regarding the class population statistics between the measurement days could be found in NB1 (Figure 8), with 92.94% class A pixels on day 1, 35.15% for day 7, and 1.00% on day 27. A similar but less pronounced effect can be seen for NB2 und NB3 in Figure 9 and Figure 10.

The spectral signatures (a) differ, dependent on the curing type. The differences in the spectral signatures of class A and B are greatest for NB1 (Figure 8) and smallest for NB3 (Figure 10). The water band in the spectral signature at 1400nm of class A is most clearly pronounced with NB1 samples (Figure 8) and becomes smaller with samples with worse curing conditions (Figure 9 and Figure 10).

Note: The dark areas on face 1 of NB1 Day 7 and Day 27 in Figure 8 resulted from contaminations after ultrasound measurements. This area was not evaluated, which had no influence on the overall results, since the area proportions of the classes are evaluated as a percentage.

The spectral signatures of the three classes (A, B, and C) are given under point (a) in the figures. The distribution of the classes for the respective curing types or on the respective measurement days is shown under point (b).

First, the combinations 1–3 are considered, where the curing types were compared with one another on measurement day 1 (Figure 5), day 7 (Figure 6), and day 27 (Figure 7).

Regarding the spectral signatures (a) in Figure 5, Figure 6 and Figure 7, a change with the age of the concrete can be observed. On the first day of measurement (Figure 5), the absorption at 1400 nm is still very pronounced, especially for class A. This amount of absorption is lower on measurement day 7 (Figure 6) and is even less on day 27 (Figure 7). On day 27, the characteristic absorption band is most pronounced around 1340 nm for both classes A and B.

Looking at the class distribution statistics (b) of the three combinations in Figure 5, Figure 6 and Figure 7, it can be determined that the NB 1 samples always had the largest proportion of class A compared to samples NB 2 and NB 3. As the curing quality deteriorates, the class A share is correspondingly lower. This effect can be observed on all measurement days. Statistics for day 1 (Figure 5) show that class A is very strongly represented, with NB1 with 90.71%. As the age of the concrete increases, the class A proportion decreases, as can be seen in Figure 6, with 53.71%, and Figure 7, with 45.97%. In the case of the NB3 samples with poor curing conditions, class B predominates on all measurement days, with 93.23% in Figure 5 (day 1), 94.15% in Figure 6 (day 7), and 68.92% in Figure 7 (day 27). The class distribution statistics from NB2 always indicate values between those of NB1 and NB3.

The influence of the lighting can be clearly seen in Figure 6; class A always dominates the left side of the samples. This effect had no influence on the overall results, since the light was the same on all measurement days and for all samples.

The results prove that differences in the hydration process according to the curing type and the duration of hydration can be determined by statistical analysis of hyperspectral data. The distribution of the corresponding classes for the samples or measurement days provides information about the quality of the hydration process and the developmental strength of concrete. Figure 11 demonstrates that the type of curing had a significant influence on the hydration of concrete. This graphic shows the class distributions of all combinations (1–6). Looking at the graphs (d)–(f) in Figure 11, the greatest change in the class distribution over the hydration days could be observed for the optimally cured samples NB 1 (d), with over 90% for class A on day 1 changing to zero until day 27. NB1 samples (d) showed a stronger development during hydration, which suggests a better development of strength. On the other hand, for NB3 (f) the class A share started with only about 80% on day 1 and was still present with about 15% on day 27, which indicates a limited development.

Figure 11a–c shows the measurement days 1, 7, and 27, where, compared to NB2 and NB3, class A always had the largest share for NB1; 90% on day 1 (a), 55% on day 7 (b), and 45% on day 27. This indicated a higher water content for optimally cured samples, which had a positive effect on the hydration process. On the other hand, compared to the other curing types, the class A proportion was lowest for the poorly cured NB3 samples, which indicated premature drying out. A low water content leads to the stopping of hydration and has a negative effect on the development of strength.

Figure 11a–c also shows that the class distribution among the three types of curing became more and more similar with increasing sample age. The biggest differences could be detected on day 1, where for NB1 class A was represented with over 90% of the pixels and for NB3 with only about 5%. The data from day 7 (b) also allow clear statements about the curing quality differences, but with increasing age it became more and more difficult to distinguish the samples from one another. On day 27 (c) the differences of the class distributions of NB1 and NB3 were less than 15%.

Differences between the curing types and measurement days can be seen in the characteristic absorption bands at 1341 nm and 1398 nm (water band). Figure 12 shows the average reflectance values for the characteristic absorption bands for class A and B of all evaluated data combinations.

The combinations 1–3 (Figure 12a–c) show the reflectance at 1341 nm and 1398 nm of class A and B after a hydration time of 1, 7, and 27 days. A reduction of absorption at 1398 nm of class A and B could be observed from day 1 (a) to day 27 (c) due to decreasing water content. The highest absorption at the water band at 1398 nm could be determined on day 1 for class A (a). With increasing age, the differences in the reflectance intensities of the two classes, A and B, decreased, which means that the spectral signatures of class A and B became more alike and the evaluation of the concrete quality became more difficult.

A similar development regarding the reflectance intensities can also be seen in the data of the different curing types in Figure 12d–f. Looking at the optimally cured samples NB1 (d), the reflectance values of classes A and B differed most strongly in the characteristic wavelength ranges and come closer with increasingly poor curing (e,f). In addition, the class A water band at 1398 nm was very pronounced with curing type NB1 (d), which was not the case for NB3 (f). This shows that the optimally cured samples had a higher water content than the poorly cured samples. In the combinations d–f, the absorption of class B at 1341 nm was always higher than the absorption of class A.

The analysis of the characteristic absorption bands of the spectral signatures allowed an evaluation of the classes and an assessment of the curing quality. A pronounced absorption at 1341 nm for class B and a clear absorption at 1398 nm for class A indicated good curing quality (Figure 12d). If the differences in reflection between the classes became smaller, as for NB3 (f), poor curing was assumed.

## 4. Conclusions and Outlook

It could be proven that the quality of concrete can be classified by comparing the hyperspectral data of samples with different curing treatments. The classification using k-means clustering enables differentiating concrete samples based on their curing quality. The samples could be analyzed in terms of their hydration process by interpreting the statistics and spectral signatures of the generated classes.

The spectra of the analyzed images showed significant changes over the hydration time. The most significant developments were determined at the absorption bands at approx. 1340 and 1400 nm.

The reflectance values for the water band at approx. 1400 nm showed that the optimally cured samples had the most extreme value and thus a higher water content compared to non-optimally cured samples. Consequently, non-optimally cured samples dry out early, which has a negative effect on the developmental strength of concrete.

The class distribution of the image pixels as a result of the k-means clustering allowed a clear distinction between the optimally, less well, and poorly cured samples. The samples could be distinguished from one another at the ages of 1, 7, and 27 days. Since the classes were divided more and more evenly among the compared samples with increasing concrete age, the tests of young concrete at the ages of 1 and 7 days gave the clearest results.

Hyperspectral analysis of young concrete enables obtaining valuable information about the hydration process, premature drying, and consequently the final quality of the concrete. It could be proven that this method is a reliable and promising application for evaluating the curing quality of concrete samples.

Further research is needed on the application of the method directly on a structure. A reliable evaluation could be realized using reference samples, whereby the characteristic absorption bands could be used as limit values for the evaluation of the curing quality. The measurement results are heavily dependent on the concrete composition, as well as on environmental influences such as light intensity and humidity. Therefore, standardized on-site tests are necessary to use the method for a reliable evaluation. It would also be worthwhile to further investigate wavelength ranges, as this would provide additional information and even clearer results.

In this respect, it is hoped that through further research in this area, the assessment of concrete curing will continue to gain in importance and that the hyperspectral method can find wide application in construction practice.

## Figures and Tables

**Figure 1 materials-14-03848-f001:**
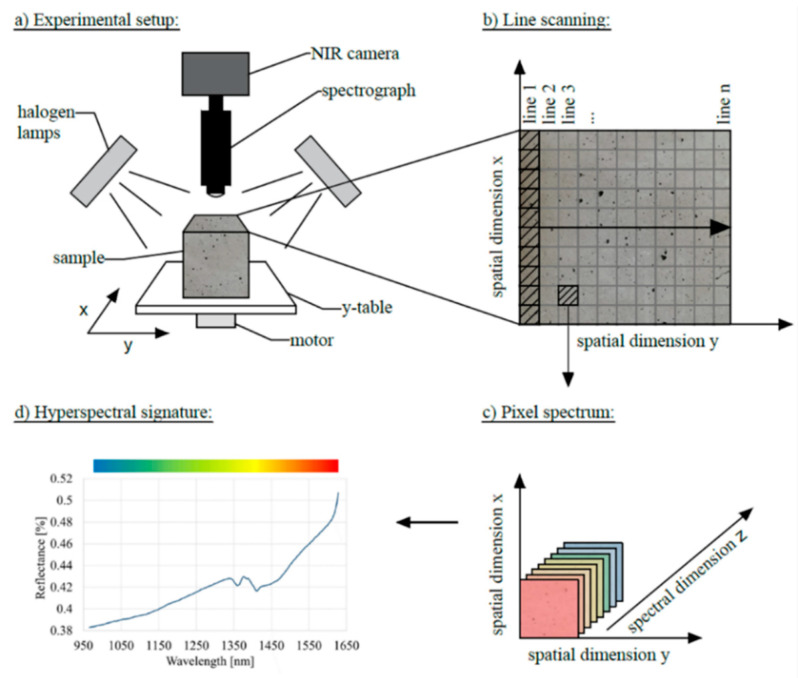
Hyperspectral imaging setup and data composition.

**Figure 2 materials-14-03848-f002:**
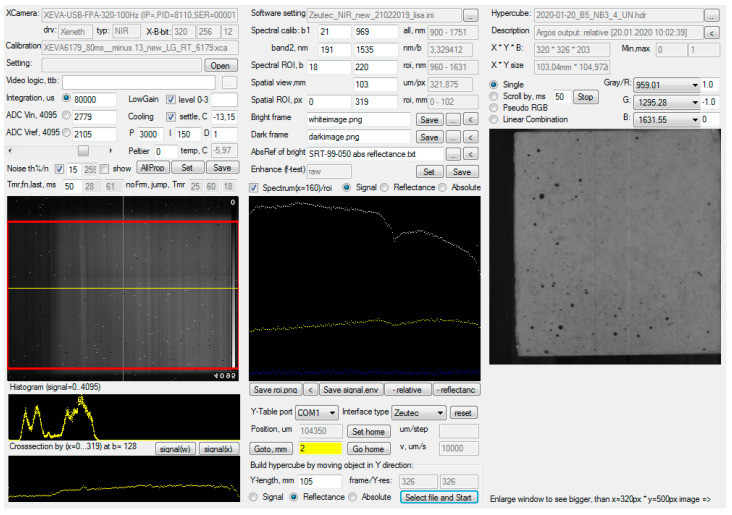
Argus software settings.

**Figure 3 materials-14-03848-f003:**
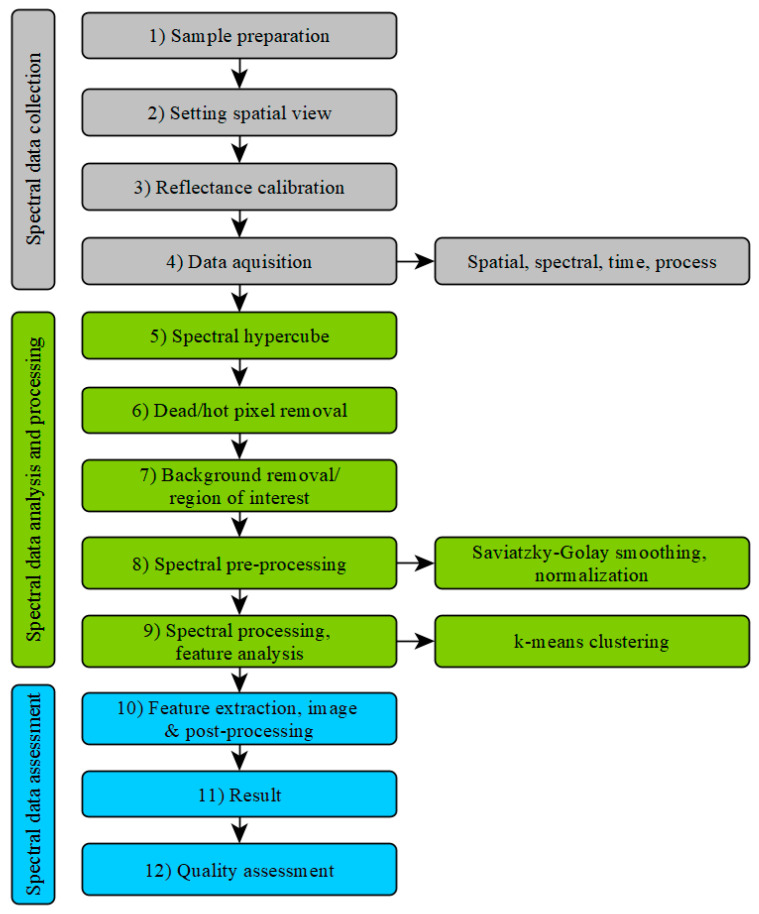
Hyperspectral imaging process steps.

**Figure 4 materials-14-03848-f004:**
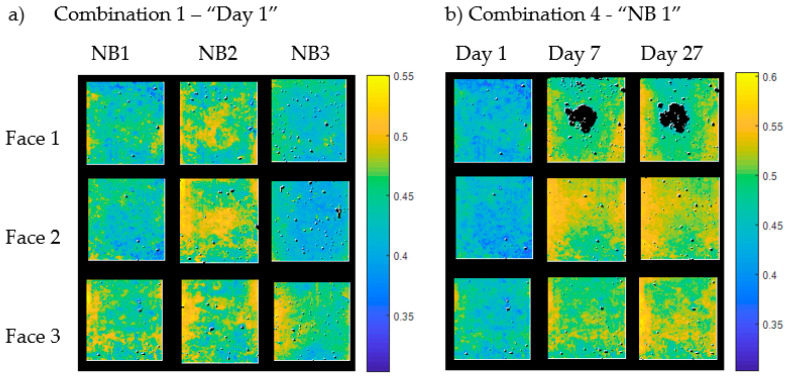
Merged images for statistical analysis of (**a**) combination 1 and (**b**) combination 4.

**Figure 5 materials-14-03848-f005:**
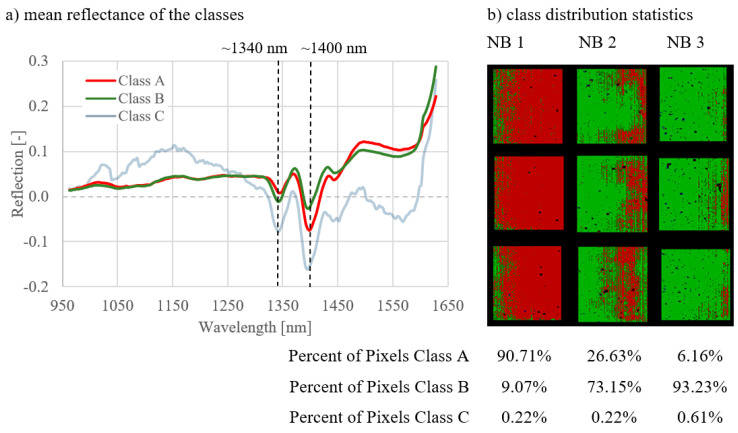
Results for combination 1; comparison of three curing types on day 1.

**Figure 6 materials-14-03848-f006:**
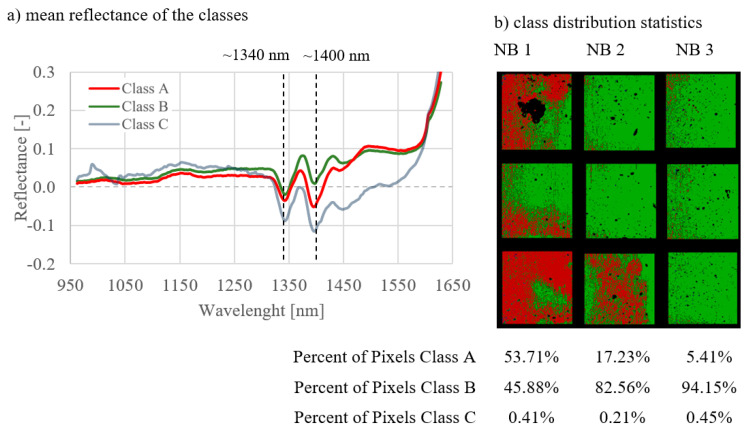
Results for combination 2; comparison of three curing types on day 7.

**Figure 7 materials-14-03848-f007:**
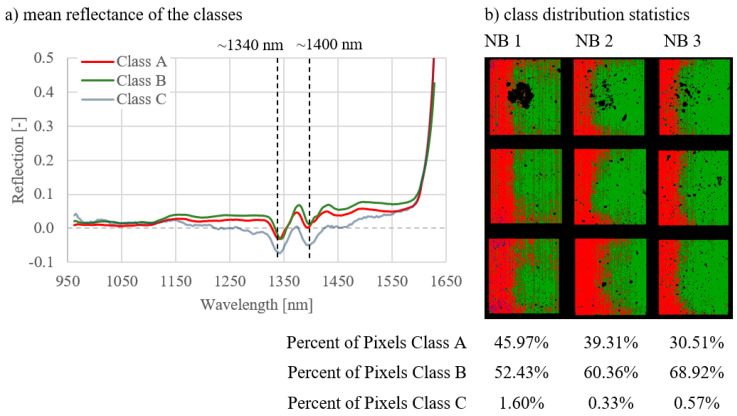
Results for combination 3; comparison of three curing types on day 27.

**Figure 8 materials-14-03848-f008:**
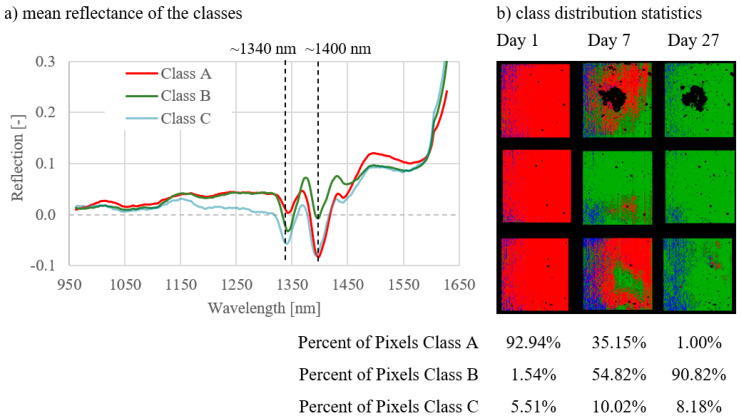
Results for combination 4; comparison of the measurement days of NB1 samples.

**Figure 9 materials-14-03848-f009:**
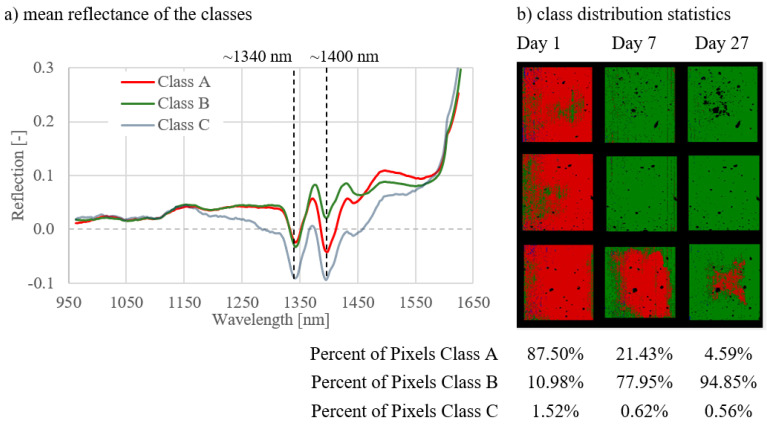
Results for combination 5; comparison of the measurement days of NB2 samples.

**Figure 10 materials-14-03848-f010:**
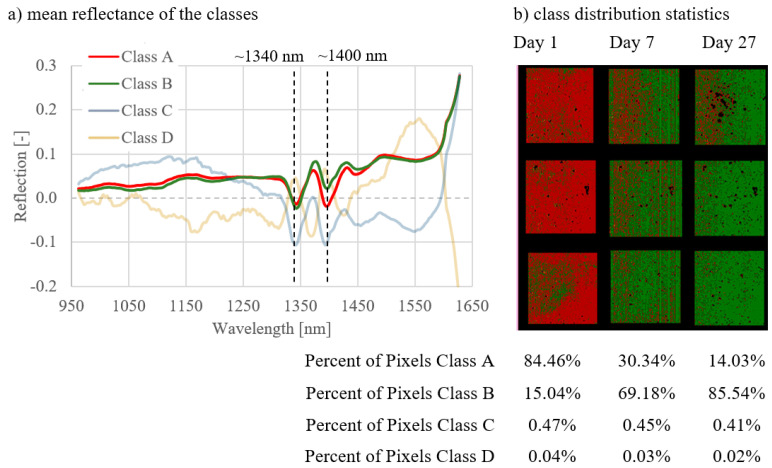
Results for combination 6; comparison of the measurement days of NB3 samples.

**Figure 11 materials-14-03848-f011:**
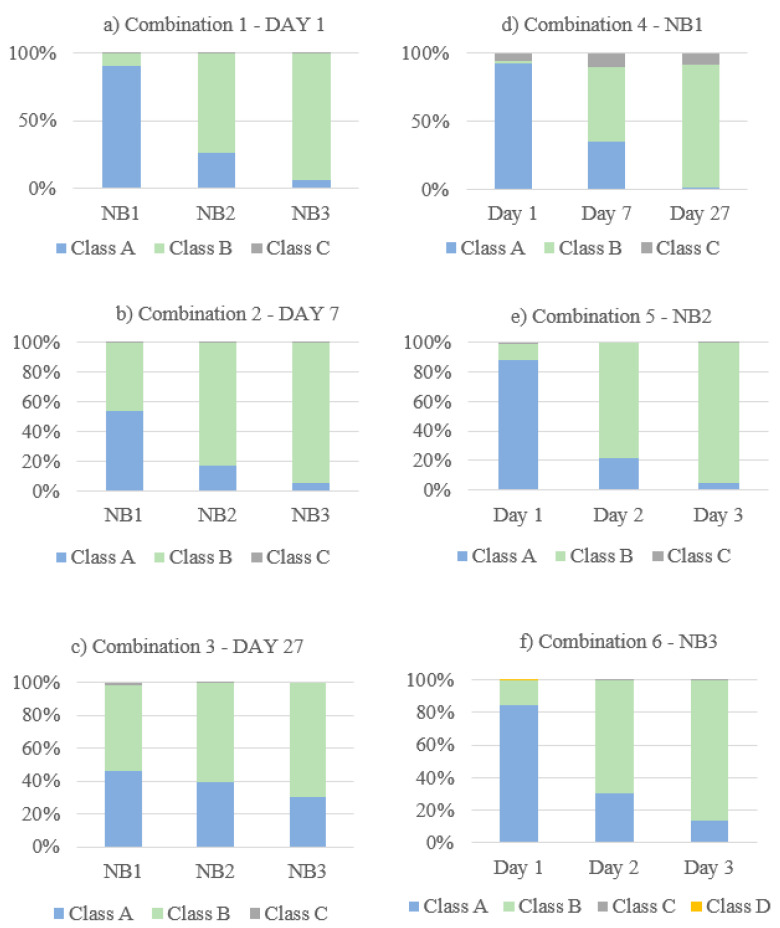
Class distribution graphs for the analyses of combinations 1–6 (**a**–**f**).

**Figure 12 materials-14-03848-f012:**
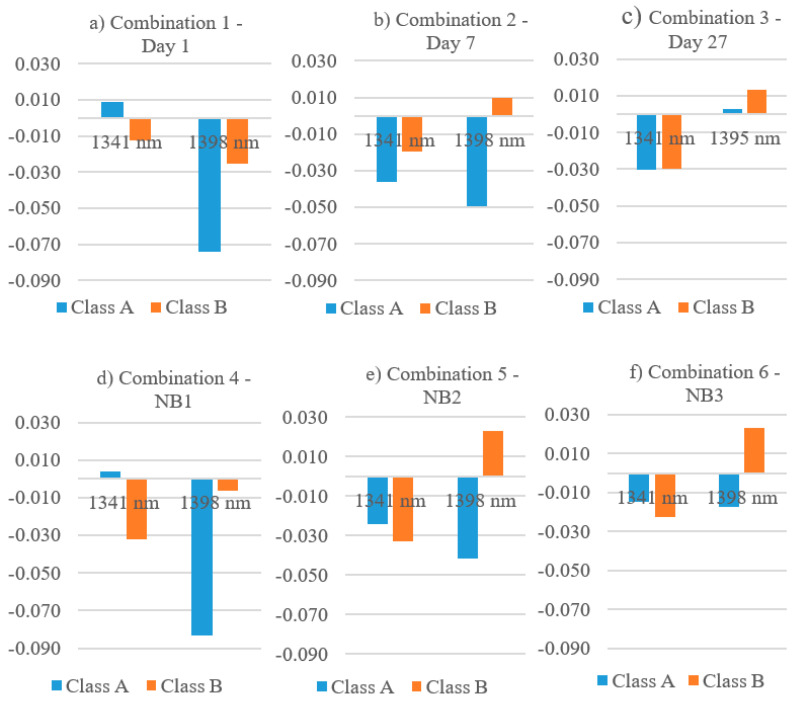
Reflectance intensities of classes A and B at 1341 and 1398 nm for combinations 1–6 (**a**–**f**).

**Table 1 materials-14-03848-t001:** Methodology and analysis of concrete quality assessment with spectral imaging from research papers.

Methodology				Pre-Processing	Analysis	Characteristic Absorption
Device	Wavelength	Resolution	Samples/Test Objects			
*The feasibility of short-wave infrared spectrometry in assessing water-to-cement (w/c) ratio and density of hardened concrete* [13]						
PerkinElmer Lambda 900 UV/VIS/NIR Spectrometer and HySpex SWIR-320m	1300–2200 nm or 1300–2500 nm	5nm or not specified	concrete specimen with different w/c ratios	standard normal variate pretreatment and Savitzky–Golay smoothing	partial least square discriminant analysis and relative percentage difference	1930 nm and 1425 nm
*Analysis of concrete reflectance characteristics using spectrometer and VNIR hyperspectral camera* [14]						
Field portable spectrometer GER-3700 and ASIA Eagle VNIR hyperspectral camera	350–2500 nm or 400–970 nm	band number: 704 or 1040	concrete specimen with different w/c ratios and different curing times, concrete structures	normalization	information extraction with ENVI software, processing in Excel	1950 nm
*Identifying the effects of different construction practices on the spectral characteristics of concrete* [15]						
Fieldspec Pro spectroradiometer	350–2500 nm	not specified	concrete samples with different treatments (control, no cure, cool cure, heat cure, …)	normalization by dividing the spectrum with the calibration spectrum (spectralon panel)	analysis of variance, analysis of the increasing of the reflectance in major regions	450 nm, 1380 nm and 1850 nm
*Reflectance spectroscopy as a tool to assess the quality of concrete* in situ [16]						
“Fieldspec Pro FRQ” VNIR-SWIR spectrometer	385–2485 nm	3700 reflectance spectra	cement pastes with different w/c ratios and curing times	normalization	logistic regression, artificial neural network	465 nm (iron oxides), 1140 nm, 1270 nm, 1450nm (hygroscopic water), *
*Assessment of Concrete Degradation with Hyper-spectral Remote Sensing* [11]						
Spectrometer GER-2600	400–2500 nm	2 nm	concrete samples exposed in carbon dioxide and solution of salt	first and second order derivative of the spectral reflectance	calculation of correlation with degradation depth, multivariant statistical analysis	440 nm, 1393 nm, 1930 nm, 2127 nm, 2340 nm
*Non-destructive chemical analysis of water and chlorine content in cement paste using near-infrared spectroscopy* [12]						
NIR-Spectrometer (FT-NIR Rocket, ARCoptix, Switzerland)	900–2600 nm	not specified	cement test pieces with different types of binders containing chloride ions	baseline and bias correction	analysis of peak wavelengths	1935 nm(water), 2257 nm (Friedel’s salt), 1412 nm (Ca (OH)_2_), 1780 nm (Ettringite)

**Table 2 materials-14-03848-t002:** Material properties of the concrete.

Austrian Concrete Type	B5
Strength class	C30/37
Cement: CEM II/A-M (S-L) 42.5 N	320 kg/m^3^
Water	170 kg/m^3^
w/c-ratio	0.48
Stone	
Round gravel 16/32 RK	451 kg/m^3^
Round gravel 8/16 RK	289 kg/m^3^
Round gravel 4/8 RK	307 kg/m^3^
Sand 0/4 RK	761 kg/m^3^
Aggregates	40 kg/m^3^
Additives	
Air entraining agent:	0.25–0.50 m%-Cement
(super)plasticizer:	0.20–0.33 m%-Cement
Air content (fresh concrete)	2.5–3.6%
Density (fresh concrete)	2383–2442 kg/m^3^
Fresh concrete temperature	21.8–23.3 °C

**Table 3 materials-14-03848-t003:** Properties and curing conditions of the samples.

Sample	Curing Conditions			w/c Ratio	Compressive Strength after 28 d Tested on 15 cm Cubes
	Storage	Temperature	Relative Humidity		
NB 1	In foil	20 °C	60%	0.48	49.7 ± 0.9 N/mm^2^
NB 2	Without foil	20 °C	60%	0.48	46.7 ± 1.2 N/mm^2^
NB 3	Without foil	30 °C	40%	0.48	38.9 ± 2.4 N/mm^2^

**Table 4 materials-14-03848-t004:** Six combinations of images were arranged to analyze the hydration behavior of differently cured samples.

Combination 1—“Day 1”			Combination 2—“Day 7”			Combination 3—“Day 27”		
NB 1	NB 2	NB 3	NB 1	NB 2	NB 3	NB 1	NB 2	NB 3
Combination 4—“NB 1”			Combination 5—“NB 2”			Combination 6—“NB 3”		
Day 1	Day 7	Day 27	Day 1	Day 7	Day 27	Day 1	Day 7	Day 27

## Data Availability

Not applicable.

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
