# Peer review of "Curing Assessment of Concrete with Hyperspectral Imaging"

_materials, 2021, doi:10.3390/ma14143848_

Round 1
Reviewer 1 Report
A very nice idea and an intensive study on the use of hypersectral imaging for the curing assessment.
The authors themselves state that further extensive investigations are required for use on the construction site. The technique seems to be very suitable for investigations on small samples.
Page 2 of 16, from line 54:
Please add that concrete curing is also favoured when constraint is removed from the structure. For example, foils can be arranged so that no large constraining stresses occur, for example, when concreting floor slabs.
All in all, a very nice article. The evaluation with k-means clustering is very interesting for the readers.
Author Response
Thank you very much for your comments!
We added the information about concrete curing in lines 51 to 53.
Reviewer 2 Report
Title: Curing assessment of concrete with hyperspectral imaging
Journal: Materials
This paper presents a new methodology approach to curing assessment of concrete.
The Authors should explain a MATLAB script.
Could the Authors implement other alternative methods to compare the results, e.g. microstructural analysis, mechanical test etc.
I recommend this work after minor corrections.
Author Response
Thank you very much for your comments! Please find our answers to your suggestions in the following:
The Authors should explain a MATLAB script: The Matlab script used for preprocessing the spectral data was developed by F. Firtha and is described in references 19 and 21. In this article, the focus was placed on the evaluation of concrete data using k-means clustering; a description of the existing Matlab scripts would go beyond the scope of the article.
Could the Authors implement other alternative methods to compare the results, e.g. microstructural analysis, mechanical test etc.:
The concrete samples as listed in Table 3 were tested in the project [20] with other non-destructive methods as gas permeability and ultrasonic testing. They showed a similar trend and confirmed the decrease in the quality of concrete with inadequately performed curing. Additionally, the compressive strength was tested on the samples as listed in Table 3.
Reviewer 3 Report
The paper, regarding the curing assessment of concrete through hyperspectral imaging brings together the most important knowledge in this field and provides a promising methodology that, after future research and standardization could become a useful nondestructive analysis instrument for concrete structures.
In the Results chapter, the discussion of figures 5-10 becomes repetitive when figures 11 and 12 are approached. The point is well made and understood without the discussion of the later two figures. The authors may decide the final structure of the paper in this regard.
Some small observations are also made, as follows:
- why the chosen curing types were 1, 7 and 27 days and not 1, 7 and 28 days? The norm is usually 28 days of curing for concrete;
- Table 1: it would be useful if the table head would be also visible on the next page;
- Table 1: the last row - "chlorine" instead of "chorine", in the title of the reference number 12;
- line 127: "and" instead of "und";
- line 151: please rephrase the caption;
- line 283: please renumber the figure as Figure 10.
Author Response
Thank you for your helpful corrections and your suggestions! Please find our answers to your comments below:
1) The authors use the Figures 11 and 12 to illustrate the effect of the different curings and measurement days by directly comparing the results of all combinations. In particular, the reflectance intensities of the characteristic wavelengths are only clearly shown in Figure 12; these cannot yet be clearly recognized in Figures 5-10. Therefore, with the approval of the reviewers, the authors would like to keep this part for better understanding.
2) Thank you for your observations in the text, we corrected all of them.
Reviewer 4 Report
Very interesting article. Research should be continued. The manuscript contains some interesting points for civil engineering researchers and practitioners.
1. The tests were carried out on cubic samples with a side of 10 cm. Is it possible to carry out tests on other samples, e.g. cubic samples with a side of 15 cm or beams?
2. Have other methods been considered in the cluster analysis?
Author Response
Thank you for your very helpful suggestions! Please find the answers to your comments below:
1) Larger objects can be measured by modification of the setup or by using e.g. portable devices. (we included this information in text line 127)
2) In the project OpitNb (see reference [20]), the results of hyperspectral imaging are compared to the results of methods as gas permeability, compressive strength, electrical resistance. In terms of concrete quality, they show the same trend as for the spectral results. The results from the other methods are not shown in this article, so as not to go beyond the scope of the article, since the focus should be on hyperspectral imaging. The cluster analysis was only performed for the hyperspectral data. The other methods required other analysis methods.